# CAUTIOUS DEEP LEARNING

## ABSTRACT

Most classifiers operate by selecting the maximum of an estimate of the conditional distribution $p(y|x)$ where $x$ stands for the features of the instance to be classified and $y$ denotes its label. This often results in a *hubristic bias*: overconfidence in the assignment of a definite label. Usually, the observations are concentrated on a small volume but the classifier provides definite predictions for the entire space. We propose constructing conformal prediction setsVovk et al. (2005) which contain a set of labels rather than a single label. These conformal prediction sets contain the true label with probability $1 - \alpha$. Our construction is based on $p(x|y)$ rather than $p(y|x)$ which results in a classifier that is very cautious: it outputs the null set — meaning "I don't know" — when the object does not resemble the training examples. An important property of our approach is that classes can be added or removed without having to retrain the classifier. We demonstrate the performance on the ImageNet ILSVRC dataset and the CelebA and IMDB-Wiki facial datasets using high dimensional features obtained from state of the art convolutional neural networks.

## 1 INTRODUCTION

We consider multiclass classification with a feature space $\mathcal{X}$ and labels $\mathcal{Y} = \{1, \dots, k\}$. Given the training data $(X_1, Y_1), \dots, (X_n, Y_n)$, the usual goal is to find a prediction function $\widehat{F} : \mathcal{X} \longmapsto \mathcal{Y}$ with low classification error $P(Y \neq \widehat{F}(X))$ where $(X, Y)$ is a new observation of an input-output pair. This type of prediction produces a definite prediction even for cases that are hard to classify.

In this paper we use conformal predictionVovk et al. (2005) where we estimate a set-valued function $C : \mathcal{X} \longmapsto 2^{\mathcal{Y}}$ with the guarantee that $P(Y \in C(X)) \geq 1 - \alpha$ for all distributions $P$. This is a distribution-free confidence guarantee. Here, $1 - \alpha$ is a user-specified confidence level. We note that the "classify with a reject option"Herbei & Wegkamp (2006) also allows set-valued predictions but does not give a confidence guarantee.

The function $C$ can sometimes output the null set. That is, $C(x) = \emptyset$ for some values of $x$. This allows us to distinguish two types of uncertainty. When $C(x)$ is a large set, there are many possible labels consistent with $x$. But when $x$ does not resemble the training data, we will get $C(x) = \emptyset$ alerting us that we have not seen examples like this so far.

There are many ways to construct conformal prediction sets. Our construction is based on finding an estimate $\widehat{p}(x|y)$ of $p(x|y)$. We then find an appropriate scalar $\widehat{t}_y$ and we set $C(x) = \{y : \widehat{p}(x|y) > \widehat{t}_y\}$. The scalars are chosen so that $P(Y \in C(X)) \geq 1 - \alpha$. We shall see that this construction works well when there is a large number of classes as is often the case in deep learning classification problems. This guarantees that $x$'s with low probability — that is regions where we have not seen training data — get classified as $\emptyset$.

An important property of this approach is that $\widehat{p}(x|y)$ can be estimated independently for each class. Therefore, $x$ is predicted to a given class in a standalone fashion which enables adding or removing classes without the need to retrain the whole classifier. In addition, we empirically demonstrate that the method we propose is applicable to large-scale high-dimensional data by applying it to the ImageNet ILSVRC dataset and the CelebA and IMDB-Wiki facial datasets using features obtained from state of the art convolutional neural networks.

**Paper Outline.** In section 2 we discuss the difference between $p(y|x)$ and $p(x|y)$. In section 3 we provide an example to enlighten our motivation. In section 4 we present the general framework of

conformal prediction and survey relevant works in the field. In section 5 we formally present our method. In section 6 we demonstrate the performance of the proposed classifier on the ImageNet challenge dataset using state of the convolutional neural networks. In section 7 we consider the problem of gender classification from facial pictures and show that even when current classifiers fails to generalize from CelebA dataset to IMDB-Wiki dataset, the proposed classifier still provides sensible results. Section 8 contains our discussion and concluding remarks. The Appendix in the supplementary material contains some technical details.

**Related Work.** There is an enormous literature on set-valued prediction. Here we only mention some of the most relevant references. The idea of conformal prediction originates from Vovk et al. (2005). There is a large followup literature due to Vovk and his colleagues which we highly recommend for the interested readers. Statistical theory for conformal methods was developed in Lei (2014); Lei et al. (2013); Lei & Wasserman (2014), and the multiclass case was studied in Sadinle et al. (2017) where the goal was to develop small prediction sets based on estimating $p(y|x)$. The authors of that paper tried to avoid outputting null sets. In this paper, we use this as a feature, similarly to Vovk et al. (2003). Finally, we mention a related but different technique called classification with the "reject option"Herbei & Wegkamp (2006). This approach permits one to sometimes refrain from providing a classification but it does not aim to give confidence guarantees.

## 2 $p(y|x)$ VERSUS $p(x|y)$

Most classifiers — including most conformal classifiers — are built by estimating $p(y|x)$. Typically one sets the predicted label of a new $x$ to be $\widehat{f}(x) = \arg\max_{y \in \mathcal{Y}} \{\widehat{p}(y|x)\}$. Since $p(y|x) = p(x|y)p(y)/p(x)$ the prediction involves the balance between $p(y)$ and $p(x|y)$. Of course, in the special case $p(y) = 1/k$ for all $y$, we have $\arg\max_{y \in \mathcal{Y}} \{p(y|x)\} = \arg\max_{y \in \mathcal{Y}} \{p(x|y)\}$.

However, for set-valued classification, $p(y|x)$ can be negatively affected by $p(y)$ and $p(x|y)$. Indeed, in this case there are significant advantages to using $p(x|y)$ to construct the classifier. Taking $p(y)$ into account ties the prediction of an observation $x$ with the likelihood of observing that class. Since there is no restriction on the number of classes, ultimately an observation should be predicted to a class regardless of the class popularity. Normalizing by $p(x)$ makes the classifier oblivious to the probability of actually observing $x$. When $p(x)$ is extremely low (an outlier), $p(y|x)$ still selects the most likely label out of all tail events. In practice this may result with most of the space classified with high probability to a handful of classes almost arbitrarily despite the fact that the classifier has been presented with virtually no information in those areas of the space. This approach might be necessary if a single class has to be selected $\forall x \in \mathcal{X}$. However, if this is not the case, then a reasonable prediction for an $x$ with small $p(x)$ is the null set.

There are also conformal methods utilizing $p(y|x)$ to predict a set of classes (Sadinle et al., 2017; Vovk et al., 2003). This methods does not overcome the inherent weakness within $p(y|x)$. As will be explained later on, the essence of this methods is to classify $x$ to $C(x) = \{y \mid P(y \mid x) \geq t\}$ for some threshold t. Due to the nature of $p(y|x)$ the points which are most likely to be predicted as the null set are when $P(y = j|x) = \frac{1}{k}$, for all classes $j \in \mathcal{Y}$. But this is exactly the points in space for which any set valued prediction should predict all class as possible.

As we shall see, conformal predictors based on $p(x|y)$ can overcome all these issues.

## 3 MOTIVATING EXAMPLE - IRIS DATASET

The Iris flower data set is a benchmark dataset often used to demonstrate classification methods. It contains four features that were measured from three different Iris species. In this example, for visualization purposes, we only use two features: the sepal and petal lengths in cm.

Figure 1 shows the decision boundaries for this problem comparing the results of (a) K-nearest neighbors (KNN), (b) support vector machines with the RBF kernel (SVM) and (c) our conformal prediction method using an estimate $\widehat{p}(x|y)$.

Both the KNN and the SVM methods provide sensible boundaries between the class where there are observations. In areas with low density $p(x)$ the decision boundaries are significantly different. The SVM classifies almost all of the space to a single class. The KNN creates an infinite strip bounded

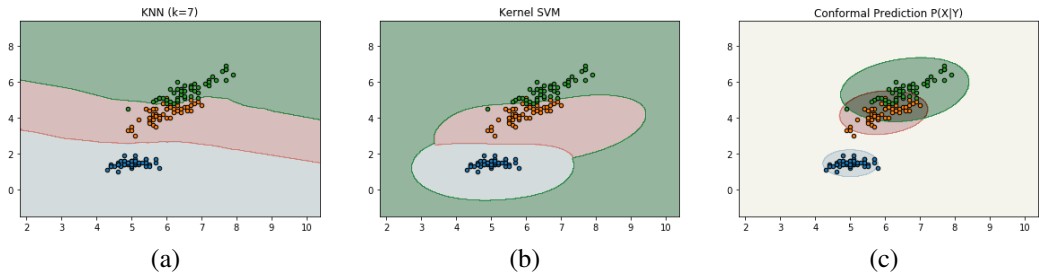

Figure 1: Classification boundaries for different methods for the Iris dataset. For the conformal prediction method (c) (with $\alpha = 0.05$) the overlapping areas are classified as multiple classes and white areas are classified as the null set. For the standard methods (a-b), the decision boundaries can change significantly with small changes in some of the data points and the prediction cannot be justified in most of the space. Online version in color.

between two (almost affine) half spaces. In a hubristic manner, both methods provide very different predictions with probability near one without sound justification.

The third plot shows the conformal set $C(x) = \{y : \ \widehat{p}(x|y) > \widehat{t}_y\}$ where the $\widehat{t}_y$ is chosen as described in Section 5. The result is a cautious prediction. If a new $X$ falls into a region with little training data then we output $\emptyset$. In such cases our proposed method modestly avoids providing any claim.

## 4 CONFORMAL PREDICTION

Let $(X_1, Y_1), \ldots, (X_n, Y_n)$ be $n$ independent and identically distributed (iid) pairs of observations from a distribution $P$. In set-valued supervised prediction, the goal is to find a set-valued function $C(x)$ such that

$$P(Y \in C(X)) \geq 1 - \alpha, \tag{1}$$

where $(X, Y)$ denotes a new pair of observations.

Conformal prediction — a method created by Vovk and collaborators Vovk et al. (2005) — provides a general approach to construct prediction sets based on the observed data without any distributional assumptions. The main idea is to construct a conformal score, which is a real-valued, permutation-invariant function $\psi(z, \mathcal{D})$ where $z = (x, y)$ and $\mathcal{D}$ denotes the training data. Next we form an augmented dataset $\mathcal{D}' = \{(X_1, Y_1), \ldots, (X_n, Y_n), (X_{n+1}, Y_{n+1})\}$ where $(X_{n+1}, Y_{n+1})$ is set equal to arbitrary values $(x, y)$. We then define $R_i = \psi((X_i, Y_i), \mathcal{D}')$ for $i = 1, \ldots, n + 1$. We test the hypothesis $H_0 : Y = y$ that the new label $Y$ is equal to $y$ using the p-value $\pi(x, y) = 1/(n+1) \sum_{i=1}^{n+1} I(R_i \geq R_{n+1})$. Then we set $C(x) = \{y : \ \pi(x, y) \geq \alpha\}$. Vovk et al. (2005) proves that $P(Y \in C(X)) \geq 1 - \alpha$ for all distributions $P$. There is a great flexibility in the choice of conformity score and 4.1 discusses important examples.

As described above, it is computationally expensive to construct $C(x)$ since we must re-compute the entire set of conformal scores for each choice of $(x, y)$. This is especially a problem in deep learning applications where training is usually expensive. One possibility for overcoming the computational burden is based on data splitting where $\widehat{p}(x|y)$ is estimated from part of the data and the conformal scores are estimated from the remaining data; see Vovk (2015); Lei & Wasserman (2014). Another approach is to construct the scores from the original data without augmentation. In this case, we no longer have the finite sample guarantee $P(Y \in C(X)) \geq 1 - \alpha$ for all distributions $P$, but we do get the asymptotic guarantee $P(Y \in C(X)) \geq 1 - \alpha - o_P(1)$ as long as some conditions are satisfied[1]. See Sadinle et al. (2017) for further discussion on this point.

### 4.1 EXAMPLES

Here are several known examples for conformal methods used on different problems.

---

[1] A sequence of random variables $X_1, X_2, \ldots,$ is $o_p(1)$ if $\forall \epsilon > 0$, $\lim_{n \to \infty} P(|X_n| \geq \epsilon) = 0$.

**Supervised Regression.** Suppose we are interested in the supervised regression problem. Let $\widehat{f} : \mathcal{X} \to \mathcal{Y}$ be any regression function learned from training data. Let $\epsilon_i$ denote the residual error of $\widehat{f}$ on the observation $i$, that is, $\epsilon_i = |\widehat{f}(X_i) - Y_i|$. Now we form the ordered residuals $\epsilon_{(1)} \leq \cdots \leq \epsilon_{(n)}$, and then define

$$C(x) = \left\{ y : |\widehat{f}(x) - y| \leq \epsilon_{(\lceil (1-\alpha) \cdot n \rceil)} \right\}.$$

If $\widehat{f}$ is a consistent estimator of $\mathbb{E}[Y|X = x]$ then $P(Y \in C(X)) = 1 - \alpha + o_P(1)$. See Lei and Wasserman Lei & Wasserman (2014).

**Unsupervised Prediction.** Suppose we observe independent and identically distributed $Y_i, \ldots, Y_n \in \mathbb{R}^d$ from distribution $P$. The goal is to construct a prediction set $C$ for new $Y$. Lei, Robins and Wasserman Lei et al. (2013) use the level set $C = \{y : \widehat{p}(y) > t\}$ where $\widehat{p}$ is a kernel density estimator. They show that if $t$ is chosen carefully then $P(Y \in C) \geq 1 - \alpha$ for all $P$.

**Multiclass Classification.** There are two notable solutions also using conformal prediction for the multiclass classification problem which are directly relevant to this work.

*Least Ambiguous Set-Valued Classifiers with Bounded Error Levels.* Sadinle et al Sadinle et al. (2017) extended the results of Lei Lei (2014) and defined $R_i = \widehat{p}(Y_i \mid X_i)$, where $\widehat{p}$ is any consistent estimator of $p(y|x)$. They defined the *minimal ambiguity* as $\mathbb{A}(C) = \mathbb{E}(|C(X)|)$ which is the expected size of the prediction set. They proved that out of all the classifiers achieving the desired $1 - \alpha$ coverage, this solution minimizes the ambiguity. In addition, the paper considers class specific coverage controlling for every class $P(Y \in C(X) \mid Y = y) \geq 1 - \alpha_y$.

*Universal Predictor.* Vovk et al Vovk et al. (2003)Sadinle et al. (2017) introduce the concept of universal predictor and provide an explicit way to construct one. A universal predictor is the classifier that produces, asymptotically, no more multiple prediction than any other classifier achieving $1 - \alpha$ level coverage. In addition, within the family of all $1 - \alpha$ classifiers that produce the minimal number of multiple predictions it also asymptotically obtains at least as many null predictions.

## 5 THE METHOD

### 5.1 THE CLASSIFIER

Let $\widehat{p}(x|y)$ be an estimate of the density $p(x|y)$ for class $Y = y$. Define $\widehat{t}_y$ to be the empirical $1 - \alpha$ quantile of the values $\{\widehat{p}(X_i|y)\}$. That is,

$$\widehat{t}_y = \sup\left\{ y : \frac{1}{n_y} \sum_i I(\widehat{p}(X_i|y) \geq t) \geq 1 - \alpha \right\} \tag{2}$$

where $n_y = \sum_i I(Y_i = y)$. Assuming that $n_y \to \infty$ and minimal conditions on $p(x|y)$ and $\widehat{p}(x|y)$, it can be shown that $\widehat{t}_y \xrightarrow{P} t_y$ where $t_y$ is the largest $t$ such that $\int_{y > t} p(x|y)dx \geq 1 - \alpha$. See Cadre et al. (2009) and Lei et al. (2013). We set $C(x) = \{y : \widehat{p}(x|y) \geq \widehat{t}_y\}$. We then have the following proposition which is proved in the appendix.

**Proposition 1** *Assume the conditions in Cadre et al. (2009) stated also in the appendix. Let $(X, Y)$ be a new observation. Then $|P(Y \in C(X)) - (1 - \alpha)| \xrightarrow{P} 0$ as $\min_y n_y \to \infty$.*

An exact, finite sample method can be obtained using data splitting. We split the training data into two parts. Construct $\widehat{p}(x|y)$ from the first part of the data. Now evaluate $\{\widehat{p}(X_i|y)\}$ on the second part of the data and define $\widehat{t}_y$ using these values. We then set $C(x) = \{y : \widehat{p}(x|y) \geq \widehat{t}_y\}$. We then have:

**Proposition 2** *Let $(X, Y)$ be a new observation. Then, for every distribution and every sample size, $P(Y \in C(X)) \geq 1 - \alpha$.*

This follows from the theory in Lei & Wasserman (2014). The advantage of the splitting approach is that there are no conditions on the distribution, and the confidence guarantee is finite sample. There is no large sample approximation. The disadvantage is that the data splitting can lead to larger prediction sets. Algorithm 1 describes the training, and Algorithm 2 describes the prediction.

---

**Algorithm 1** Training Algorithm

---

**Input:** Training data $Z = (X, Y)$, Class list $\mathcal{Y}$, Confidence level $\alpha$, Ratio $p$.
$\widehat{p}_{list} = list; \widehat{t}_{list} = list$                                    ▷ Initialize lists
**for** $y$ in $\mathcal{Y}$ **do**                          ▷ Loop over all the classes independently
     $X_{tr}^y, X_{val}^y \leftarrow SubsetData\left(Z, \mathcal{Y}, p\right)$                  ▷ Split $X \mid y$ with ratio $p$
     $\widehat{p}_y \leftarrow LearnDensityEstimator\left(X_{tr}^y\right)$
     $\widehat{t}_y \leftarrow Quantile\left(\widehat{p}_y\left(X_{val}^y\right), \alpha\right)$              ▷ The validation set $\alpha$ quantile
     $\widehat{p}_{list}.append\left(\widehat{p}_y\right); \widehat{t}_{list}.append\left(\widehat{t}_y\right)$
**return** $\widehat{p}_{list}; \widehat{t}_{list}$

---

**Algorithm 2** Prediction Algorithm

---

**Input:** Input to be predicted $x$, Trained $\widehat{p}_{list}; \widehat{t}_{list}$, Class list $\mathcal{Y}$.
$C = list$                                            ▷ Initialize $C\left(x\right)$
**for** $y$ in $\mathcal{Y}$ **do**                        ▷ Loop over all the classes independently
     **if** $\widehat{p}_y\left(x\right) \geq \widehat{t}_y$ **then**
         $C.append\left(y\right)$
**return** $C$

---

## 5.2 CLASS ADAPTIVITY

As algorithms 1 and 2 demonstrate, the training and prediction of each class is independent from all other classes. This makes the method adaptive to addition and removal of classes ad-hoc. Intuitively speaking, if there is $1 - \alpha$ probability for the observation to be generated from the class it will be classified to the class regardless of any other information.

Another desirable property of the method is that it is possible to obtain different coverage levels per class if the task requires that. This is achieved by setting $\widehat{t}_y$ to be the $1 - \alpha_y$ quantile of the values $\{\widehat{p}(X_i|y)\}$.

## 5.3 DENSITY ESTIMATION

The density $p(x|y)$ has to be estimated from data. We use the standard kernel density estimation method, which was shown to be optimal in the conformal setting under weak conditions in Lei, Robins and Wasserman Lei et al. (2013).

Density estimation in high dimensions is a difficult problem. Nonetheless, as we will show in the numerical experiments (Section 6), the proposed method works well in these tasks as well. An intuitive reason for this could be that the accuracy of the conformal prediction does not actually require $\widehat{p}(x|y)$ to be close to $p(x|y)$ in $L_2$. Rather, all we need is that the ordering imposed by $\widehat{p}(x|y)$ approximates the ordering defined by $p(x|y)$. Specifically, we only need that $\{(x, x') : p(x|y) > p(x'|y) + \Delta\}$ is approximated by $\{(x, x') : \widehat{p}(x|y) > \widehat{p}(x'|y) + \Delta\}$ for $\Delta > 0$. We call this "ordering consistency." This is much weaker than the usual requirement that $\int (\widehat{p}(x|y) - p(x|y))^2 dx$ be small. This new definition and implications on the approximation of $p(x \mid y)$ will be further expanded in future work.

## 6 IMAGENET CHALLENGE EXAMPLE

The ImageNet Large Scale Visual Recognition Challenge (ILSVRC) Deng et al. (2009) is a large visual dataset of more than $1.2$ million images labeled across $1,000$ different classes. It is considered a large scale complex visual dataset that reflects object recognition state-of-the-art through a yearly competition.

In this example we apply our conformal image classification method to the ImageNet dataset. We remove the last layer from the pretrained Xception convolutional neural network Chollet (2016) and use it as a feature extractor. Each image is represented as a $2,048$ dimensional feature in $\mathbb{R}^{2048}$. We learn for each of the $1,000$ classes a unique kernel density estimator trained only on images within the training set of the given class. When we evaluate results of standard methods we use the

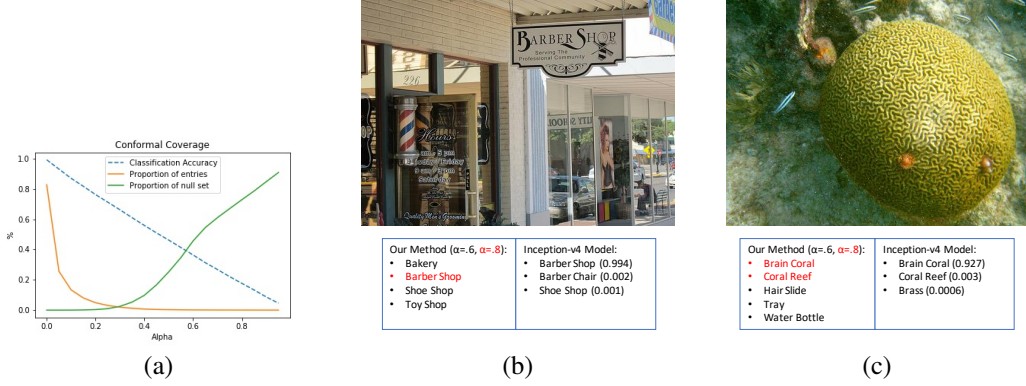

Figure 2: (a) Performance plot for the conformal method. Accuracy is empirically linear as a function of $\alpha$ but affect the number of classes predicted per sample. (b-c) are illustrative examples. When $\alpha = 0.6$ both black and red classes are predicted. When $\alpha = 0.8$ the red classes remain.

Inception-v4 model Szegedy et al. (2017) to avoid correlation between the feature extractor and the prediction outcome as much as possible.

The Xception model obtains near state-of-the-art results of $0.79$ (top-1) and $0.945$ (top-5) accuracy on ImageNet validation set. As a sanity check to the performance of our method, selecting for each image the highest (and top 5) prediction of $\widehat{p}(x \mid y)$ achieves $0.721$ (top-1) and $0.863$ (top-5) on ImageNet validation set. We were pleasantly surprised by this result. Each of the $\widehat{p}(x \mid y)$'s were learned independently possibly discarding relevant information on the relation between the classes. The kernel density estimation is done in $\mathbb{R}^{2,048}$ and the default bandwidth levels were used to avoid overfitting the training set. Yet the naive performance is roughly on par with GoogLeNet Szegedy et al. (2015) the winners of $2014$ challenge (top-1: $0.687$, top-5: $0.889$).

For conformal methods the confidence level is predefined. The method calibrates the number of classes in the prediction sets to satisfy the desired accuracy level. The the main component affecting the results is the hyperparameter $\alpha$. For small values of $\alpha$ the accuracy will be high but so does the number of classes predicted for every observation. For large values of $\alpha$ more observations are predicted as the null set and less observations predicted per class. Figure 2 (a) presents the trade-off between the $\alpha$ level and the number of classes and the proportion of null set predictions for this example. For example $0.5$, accuracy would require on average $2.7$ predictions per observation and $0.252$ null set predictions. The actual selection of the proper $\alpha$ value is highly dependent on the task. As discussed earlier, a separate $\alpha_y$ for each class can also be used to obtain different accuracy per class.

Figures 2 (b) and (c) show illustrative results from the ImageNet validation set. (b) presents a picture of a "*Barber Shop*". When $\alpha = 0.6$ the method correctly suggests the right class in addition to several other relevant outcomes such as "*Bakery*". When $\alpha = 0.8$ only the "*Barber Shop*" remains. (c) show a "*Brain Coral*". For $\alpha = 0.6$ the method still suggests classes which are clearly wrong. As $\alpha$ increases the number of classes decrease and for $\alpha = 0.8$ only "*Brain Coral*" and "*Coral Reef*" remains, both which are relevant. At $\alpha = 0.9$ "*Coral Reef*" remains, which represents a misclassification following from the fact that the class threshold is lower than that of "*Brain Coral*". Eventually at $\alpha = 0.95$ the null set is predicted for this picture.

Figure 5 shows a collage of 20 images using $\alpha = 0.7$. To avoid selection bias we've selected the first 20 images in the ImageNet validation set.

## 6.1 OUTLIERS

Figure 3 (a) shows the outcome when the input is random noise. We set the threshold $\alpha = 0.01$. This gives a less conservative classifier that should have the largest amount of false positives. Even with such a low threshold all $100$ random noise images over $1,000$ categories are correctly flagged as

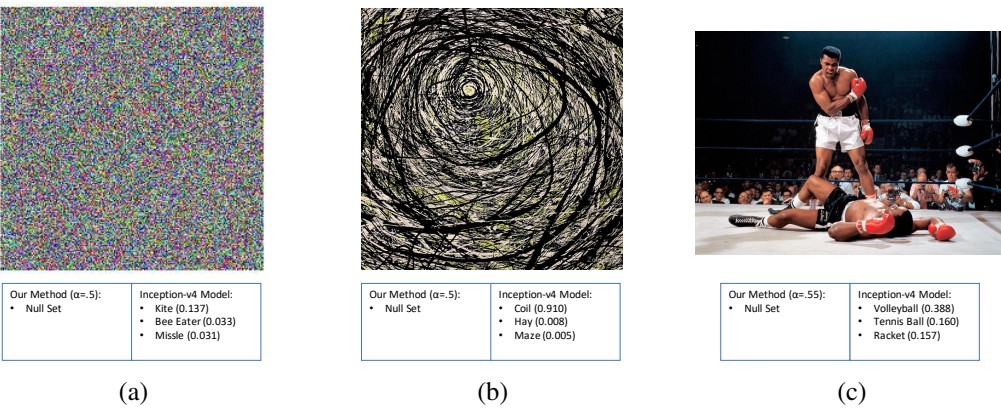

Figure 3: Classification results for (a) random noise; (b) Jackson Pollock "Rabit Hole"; (c) Muhammad Ali towering over Sonny Liston (1965 rematch). These pictures are outliers for the Imagenet categories. The left labels of each picture are provided by our method and the right are the results of the Inception-v4 model Szegedy et al. (2017).

the null set. Evaluating the same sample on the Inception-v4 model Szegedy et al. (2017) results with a top prediction average of $0.0836$ (with $0.028$ standard error) to "Kite" and $0.0314$ ($0.009$) to "Envelope". The top-5 classes together has mean probability of $0.196$, much higher than the uniform distribution expected for prediction of random noise.

Figure 3 (b) show results on Jackson Pollock paintings - an abstract yet more structured dataset. Testing 11 different paintings with $\alpha = 0.5$ all result with the null set. When testing the Inception-v4 model output, $7/11$ paintings are classified with probability greater than $0.5$ to either "Coil", "Ant", "Poncho", "Spider Web" and "Rapeseed" depending on the image.

Figure 3 (c) is the famous picture of Muhammad Ali knocking out Sonny Liston during the first round of the 1965 rematch. "Boxing" is not included within in the ImageNet challenge. Our method correctly chooses the null set with $\alpha$ as low as $0.55$. Standard method are forced to associate this image with one of the classes and choose "Volleyball" with $0.38$ probability and the top-5 are all sport related predictions with $0.781$ probability. This is good result given the constraint of selecting a single class, but demonstrate the impossibility of trying to create classes for all topics.

## 7    GENDER RECOGNITION EXAMPLE

In the next example we study the problem of gender classification from facial pictures. CelebFaces Attributes Dataset (CelebA) Liu et al. (2015) is a large-scale face attributes dataset with more than $200K$ celebrity images attributed, each with 40 attribute annotations including the gender (Male/Female). IMDB-Wiki dataset is a similar large scale ($500K+$ images) dataset Rothe et al. (2016) with images taken from IMDB and Wikipedia.

We train a standard convolutional neural network (5 convolution and 2 dense layers with the corresponding pooling and activation layers) to perform gender classification on CelebA. It converges well obtaining $0.963$ accuracy on a held out test set, but fails to generalize to the IMDB-Wiki dataset achieving $0.577$ accuracy, slightly better than a random guess. The discrepancy between the two datasets follows from the fact that facial images are reliant on preprocessing to standardize the input. We have used the default preprocessing provided by the datasets, to reflect a scenario in which the distribution of the samples changes between the training and the testing. Figure 4 (a) and (b) show mean pixel values for females pictures within CelebA vs pictures in the IMDB-Wiki dataset. As seen, the IMDB-Wiki is richer and offers larger variety of human postures.

Although the standard classification method fails in this scenario, the conformal method suggested in this paper still offers valid and sensible results both on CelebA and IMDB-Wiki when using the features extracted from the network trained on CelebA. Figure 4 (c) shows the performance of the method with respect to both dataset. CelebA results are good since they are based on features that

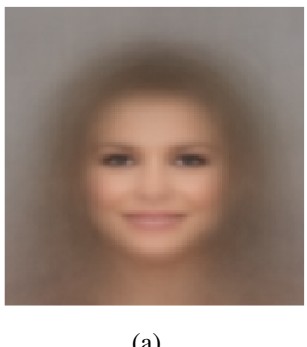 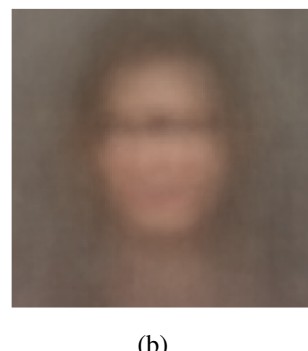 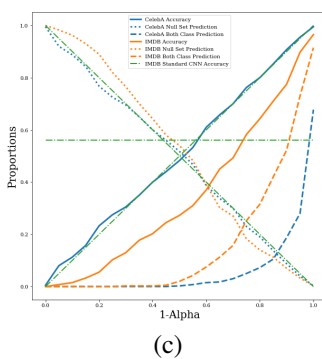

(a)           (b)           (c)

Figure 4: (a) Females faces mean pixel values in (a) CelebA; (b) IMDB-Wiki. Within CelebA the pictures are aliened with fixed posture, explaining why it naively fails to generalize to IMDB-Wiki images. (c) Performance plots for the conformal method on both CelebA and IMDB-Wiki.

perform well for this dataset. The level of accuracy is roughly $1 - \alpha$ as expected by the design, while the proportion of null predictions is roughly $\alpha$. Therefore for all $\alpha$ there are almost no false positives and all of the errors are the null set.

The IMDB-Wiki results are not as good, but better than naively using a $0.577$ accuracy classifier. Figure 4 (c) show the classifier performance as a function of $\alpha$. Both the accuracy and the number of false positives are tunable. For high values of $1-\alpha$ the accuracy is much higher than $0.577$, but would results in a large number of observations predicted as both genders. If cautious and conservative prediction is required small values of $1 - \alpha$ would guarantee smaller number of false predictions, but a large number of null predictions. The suggested conformal method provide a hyper-parameter controlling which type of errors are created according to the prediction needs, and works even in cases where standard methods fail.

## 8  Discussion

In this paper we showed that conformal, set-valued predictors based on $\widehat{p}(x|y)$ have very good properties. We obtain a cautious prediction associating an observation with a class only if the there is high probability of that observation is generated from the class. In most of the space the classifier predicts the null set. This stands in contrast to standard solutions which provide confident predictions for the entire space based on data observed from a small area. This can be useful when a large number of outliers are expected or in which the distribution of the training data won't fully describe the distribution of the observations when deployed. Examples of such, are object recognition systems used online continuously in real life. We also obtain a large set of labels in the set when the object is ambiguous and is consistent with many different classes. Thus, our method quantifies two types of uncertainty: ambiguity with respect to the given classes and outlyingness with respect to the given classes.

In addition, the conformal framework provides our method with its coverage guarantees and class adaptivity. It is straightforward to add and remove classes at any stage of the process while controlling either the overall or class specific coverage level of the method in a highly flexible manner. This desired properties comes with a price. The distribution of $p(x|y)$ for each class is learned independently and the decision boundaries are indifferent to data not within the class. This might lead to decision boundaries which are inferior to current methods if the only goal is to distinguish between classes which are fully described by the training data. Alternative methods to learn $p(x|y)$ taking all the training data into account might overcome this limitation and can be the focus of future investigations.

During the deployment of the method, evaluation of a large number of kernel density estimators is required. This is relatively slow compared to current methods. This issue can be addressed in future research with more efficient ways to learn ordering-consistent approximations of $p(x|y)$ that can be deployed on GPU's.

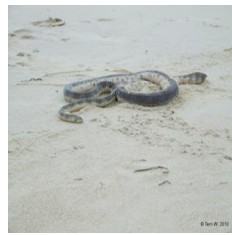
TL: Sea Snake
Prediction: Null Set

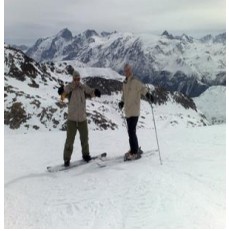
TL: Alp
Prediction: Ski

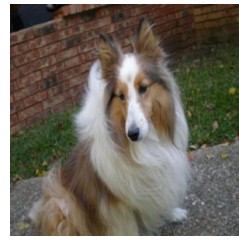
TL: Shetland Sheepdog
Prediction: Shetland
Sheepdog, Collie, Toilet
Paper

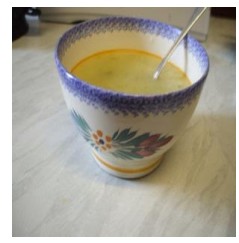
TL: Soup Bowl
Prediction: Face
Powder, Soup Bowl,
Tray

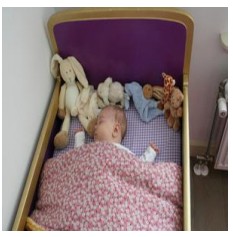
TL: Cradle
Prediction: Sleeping
Bag

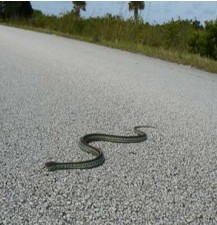
TL: Garter Snake
Prediction: Null Set

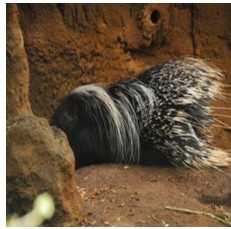
TL: Porcupine
Prediction:
Porcupine, Quill

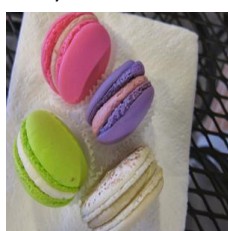
TL: Bakery
Prediction: Null Set

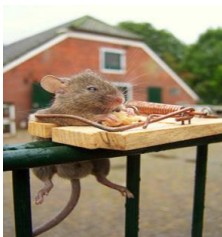
TL: Mousetrap
Prediction: Mousetrap

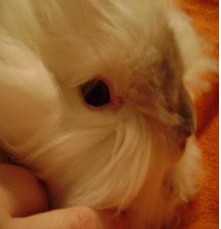
TL: Angora
Prediction: Null Set

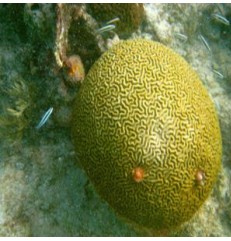
TL: Brain Coral
Prediction: Brain Coral,
Water Bottle, Coral
Reef

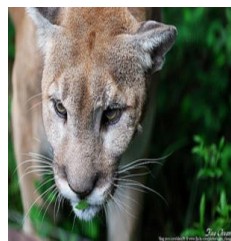
TL: Cougar
Prediction: Cougar

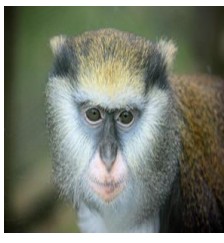
TL: Guenon
Prediction: Guenon,
Patas

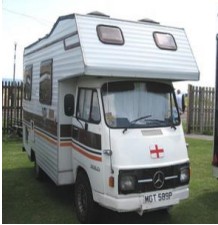
TL: Recreational Vehicle
Prediction: Recreational
Vehicle

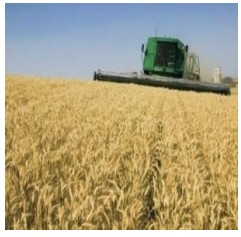
TL: Harvester
Prediction: Null Set

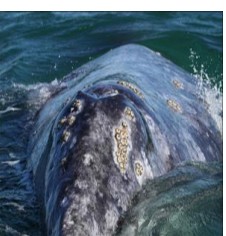
TL: Grey Whale
Prediction: Null Set

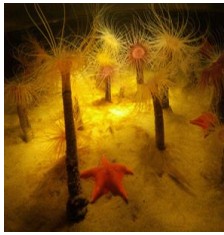
TL: Sea Anemone
Prediction: Null Set

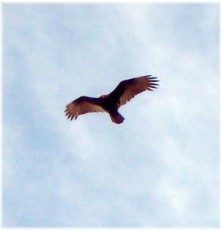
TL: Vulture
Prediction: Null Set

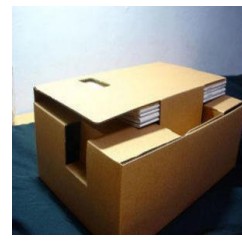
TL: Carton
Prediction: Null Set

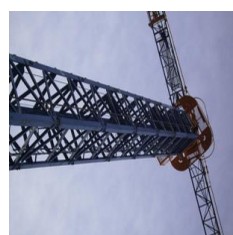
TL: Crane
Prediction: Crane,
Hook

Figure 5: A collage of the first 20 images in the ImageNet validation set with $\alpha = 0.7$. *TL* denotes the image true label and *Prediction* is the method output. By design only $0.3$ accuracy is expected, yet both the true and the false predictions are reasonable. Online version in color.

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

APPENDIX: DETAILS ON PROPOSITION 1

Here we provide more details on Proposition 1. We assume that the conditions in Cadre et al. (2009) hold. In particular, we assume that $nh_y^d/(\log n)^{16} \to \infty$ and $nh_y^{d+4}(\log n)^2 \to 0$ where $h_y$ is the bandwidth of the density estimator. In addition we assume that $\mathcal{X}$ is compact and that $\min_y n_y \to \infty$ where $n_y = \sum I(Y_i = y)$.

Let $C(x) = \{y : p(x|y) > t_y\}$ and $\widehat{C}(x) = \{y : \widehat{p}(x|y) > t_y\}$. Note that, conditional on the training data $\mathcal{D}$,

$$P(Y \in \widehat{C}(X)) = \sum_y \int I(y \in \widehat{C}(x))p(x|y)dx$$

$$= \sum_y \int I(y \in \widehat{C}(x))\widehat{p}(x|y)dx + \sum_y \int I(y \in \widehat{C}(x))[p(x|y) - \widehat{p}(x|y)]dx.$$

From Theorem 2.3 of Cadre et al. (2009) we have that $\mu(\{p(x|y) \geq t\}\Delta\{p(x|y) \geq \widehat{t}\}) = O_P(\sqrt{1/(n_y h_y^d)}) = o_P(1)$ where $\mu$ is Lebesgue measure and $\Delta$ denotes the set difference. It follows that

$$\int I(y \in \widehat{C}(x))p(x|y)dx = \int I(y \in C(x))p(x|y)dx + o_P(1) = 1 - \alpha + o_P(1).$$

Also,

$$\left| \sum_y \int I(y \in \widehat{C}(x))[p(x|y) - \widehat{p}(x|y)]dx \right| \leq \sum_y \int I(y \in \widehat{C}(x))|p(x|y) - \widehat{p}(x|y)|dx$$

$$\leq k \max_y ||\widehat{p}(x|y) - p(x|y)||_\infty \xrightarrow{P} 0$$

since, under the conditions, $\widehat{p}(x|y)$ is consistent in the $\ell_\infty$ norm. It follows that $P(Y \in \widehat{C}(X)) = 1 - \alpha + o_P(1)$ as required.

We should remark that, in the above, we assumed that the number of classes is fixed. If we allow $k$ to grow the analysis has to change. Summing the errors in the expression above we have that $P(Y \in \widehat{C}(X)) = 1 - \alpha + R$ where now the remainder is

$$R = O\left( \sum_y \frac{1}{\sqrt{n_h h_h^d}} + \sum_y \left( \frac{\log n_y}{n_y} \right)^{\frac{2}{4+d}} \right).$$

We then need assume that as $k$ increases, the $n_y$ grow fast enough so that $R \xrightarrow{P} 0$. However, this condition can be weakened by insisting that for all $y$ with $n_y$ small, we force $\widehat{C}$ to omit $y$. If this is done carefully, then the coverage condition can be preserved and we only need $R$ to be small when summing over the larger classes. The details of the theory in this case will be reported in future work.

