# OpenReview forum: "Cautious Deep Learning"
_ICLR.cc/2019/Conference_

### Official Review · AnonReviewer3 · 2018-10-29
**Interesting for the ICLR community, but somewhat straightforward**

**Rating:** 4
**Confidence:** 5

**Review:**

The paper proposes deep learning extension of the classic paradigm of 'conformal prediction'. Conformal prediction is similar to multi-label classification, but with a statistical sound way of thresholding each (class-specific) classifier: if our confidence in the assignment of an x to a class y is smaller than \alpha, then we say 'do not know / cannot classify'). This is interesting when we expect out of distribution samples (e.g., adversarial ones).

I think this paper, which is very well written, would make for nice discussions at ICLR, because it is (to my knowledge) the first that presents a deep implementation of the conformal prediction paradigm.  However, there are a couple of issues, which is why I think it is definitely not a must have at ICLR. The concrete, deep implementation of the approach is rather straightforward and substandard for ICLR: Features are taken from an existing, trained SOTA DNN, then input KDE, based on which for each class the quantiles are computed (using a validation set). Thus, feature and hypothesis learning are not coupled, and the approach requires quite a lot of samples per class (however, oftentimes in multi-label prediction we observe a Zipf law, ie many classes have fewer than five examples). Furthermore, there is no coupling between the classes; each class is learned separately; very unlikely this will work better than a properly trained multi-class or (e.g., one-vs.-rest) multi-label classifier in practice. Since a validation set is used to compute the quantiles, substantial 'power' is lost (data not used very efficiently; although that could be improved at the expense of expensive CV procedures).

---

> ### Author Response · Authors · 2018-11-16
> **The novelity is not with the features extractions**
>
> The main point was addressed in the general response.
> Regarding the other issues:
> “Fewer than 5 ” - we agree that our method requires at least few hundreds of observation per class, and fewer than 5 will be an issue. But that's an issue with many other methods, and is great to have in a new method, not a criteria for publication.
> “No coupling between the classes” - we think that in some scenarios this might be a feature, not an problem. For example, when parallelization is desired, or dynamic addition/removal of classes without retraining. But we agree with the reviewer that if all that needed is to separate between the classes, this would lead to inferior results in comparison to standard methods. We are currently studying a way to create a regularization between the classes. Specifically, penalize points which obtain high density with multiple classes at the same time, and encourage the selection of high density points only on a single class. Reviewer can see results on the Iris dataset here:
> https://www.dropbox.com/s/b4e0jawu1ozlo21/Alpha_95_Neighbors_5_Gamma_-0.50.png?dl=0
> This output decision boundaries which closely resembles standard methods, while still obtaining the conformal cautiousness. We intend to fully address this point in the final version.

---

### Official Review · AnonReviewer2 · 2018-11-04
**Conformal Methods**

**Rating:** 7
**Confidence:** 2

**Review:**

This paper applies Conformal Methods to multi-class classification. I am unfamiliar with the field, but the authors seem to be the first to attempt multiclass classification with Conformal Methods by estimating p(x|y) instead of the usual p(y|x). In doing so, they effectively build an independent classifier for each class that estimates whether an example comes from that class within a certain confidence which is set before training time.
In this way, they create meaning NULL predictions for an unidentified example, instead of the usual low-probability of an erroneous class.
The paper is well written, although it is difficult for me to work out which parts are the author's contributions, and which parts are tutorial/introduction to known Conformal Methods techniques, again this might be because this is not my subject area.
The ImageNet results look OK, not great, I would prefer to see a table in section 6.
The transfer of features from CelebA to IMDB-wiki is good, but it is hard to tell how good. I feel there should be more comparisons to other methods, even, perhaps an ROC curve for a set of 1 vs all mlp binary/SVM classifiers along with the conformal results (mlp using different cutoff thresholds, conformal method being trained with different confidence levels).
I feel like this paper may be important, but it is a little difficult for me (myself) to judge without clear empirical evidence of a strong method vs other methods.

---

> ### Author Response · Authors · 2018-11-16
> **Regarding the Novelity**
>
> We thank you for the review.
>
> Regarding the novelty vs tutorial/introduction to known conformal methods - it is known that when selecting 1-alpha of the joint density, it is possible to obtain conformal predictors that achieve the desired coverage.
>
> The novelity of this paper is to say that you can take the 1-alpha coming from the P(x|y) direction instead of P(y|x) direction. That leads to a whole different structure of results, that makes more conceptual sense to use. Reviewer is encouraged to compare the iris example results with the non-intuitive shapes suggested at "Least Ambiguous Set-Valued Classifiers with Bounded Error Levels" when using P(y|x).

---

### Official Review · AnonReviewer1 · 2018-11-05
**More quantitative evaluations are necessary**

**Rating:** 4
**Confidence:** 3

**Review:**

The paper proposes an approach to construct conformal prediction sets. The idea is to estimate p(x|y) and then construct a conformal set based on the p(x|y) instead of p(y|x). The paper claims that such a method produces cautious classifiers that can produce "I don't know" answers in the face of uncertainty.

However,
A] Although the paper is titled is "Cautious Deep Learning", the method is broadly applicable to any classifier, there is nothing in the method that restricts it to the domain of deep learning. A broad spectrum evaluation could have been done on standard multi-class classifiers.
B] The paper provides multiple qualitative evaluation results. While it gets the point across, I still would have liked to see a quantitative evaluation, for e.g., there have been several papers that proposed generating adversarial examples for deep learning. The author could have taken any of those methods, generated adversarial examples for deep learning and compared the original classifier with the conformal prediction set. Also, such comparison would have made the paper more connected with deep learning.
C] The paper uses Xception network as a feature extractor and then compares its result with Inception-v4. Honestly, I would have preferred if the comparison was between 1] Xception Feature Extractor + Conformal Set Prediction, 2] Xception network prediction, and 3] Inception-v4. The reason being that it is very difficult to understand how much of the cautious-ness is because of the proposed approach and how much is due to the Xception network being good. For example, in Figure 3b, does Xception network generate high probability values for the top classes or does it generate low probability values? Unless we can understand this difference, it is very difficult to appreciate what this approach is giving us.
D] Another analysis that could have been done is to apply this approach and use several different pre-trained networks as feature extractor and check whether there is a decrease in false positives across all the networks, that would suggest that the method can truly make deep learning cautious across a wide variety of networks.
E] Another analysis that could have been done is understand the impact of the quality of feature extractor. For example, take a deep network (of sufficient depth) and use the proposed approach but instead of using just the penultimate layer for feature extraction, one can keep on removing layers from the end and use the remaining as the feature extractor. Then analyze the quality of conformal predictions as each layer gets removed. One can understand the robustness of this method.

Even though doing all these evaluations may be a tad too much, but definitely, quite a few of those could have been done to make the approach look convincing and enticing. I think bulking this paper with such analysis could make for a very good submission. However, as it stands, it still quite lacks.

---

> ### Author Response · Authors · 2018-11-16
> **It is not "apples to apples"**
>
> A] We agree, but consider that as a feature, not an issue. This will be stated more general in future versions of this paper. Not sure how you can report results on visual data without using CNNs. An evaluation of standard multi-class classifiers on standard data would show that the better the features the more conservative the prediction is (less classes predicted per observation), as also demonstrated in the gender recognition example.
> B, C, D and E] The main point we want to iterate is that softmax has inherent weakness by design and that it's possible to report results alternatively. The suggestions (which are interesting and valuable) focus on studying the effect of the features used. We believe current experiments better exemplify the method contribution. In future research we will be happy to study the effect of features, but we are already at 9 pages + appendix in a concisely written paper.
>
> We would like to point out that the application of a conceptually new (and highly general) approach straightaway on ImageNet obtaining the exact theoretical guarantees is highly non-trivial.

---

> > ### Comment · AnonReviewer1 · 2018-11-29
> > **It is indeed not "apples to apples"**
> >
> > >> We believe current experiments better exemplify the method contribution.
> >
> > This is precisely the problem I am having. The experiments are anecdotal (except the gender recognition experiment) and competing methods are working on top of different feature sets. This creates a question in my mind as to whether the proposed approach can really work around the "inherent weakness" of the softmax. This is why I proposed analysis C above.

---

### Author Response · Authors · 2018-11-16
**The contribution is a different way to report results from features**

Most Neural Network architectures can be considered a composition of two components. (1) Feature extractor that is built from many different layers and learn representations of the input; (2) A softmax layer performing logistic regression to provide the prediction.
The majority of the published papers focus on (1) (new layers, different architectures, hyper-parameter tuning, etc). Usually (2) is taken for granted in most solutions. Our paper focus on (2).
Reviewer 1 and 3 which object our contribution judge it as if it was related to (1). Reviewer 3 point out that our deep learning usage of (1) is “straightforward and substandard” (we agree) and reviewer 1 suggest we study how the different features effect the results.
Both reviewers disregard the fact that our contribution is aimed to provide an alternative to (2). We claim that the usage of softmax contains a normalization component which leads to over confident predictions throughout the space without any empirical data to support this hubristic predictions. We suggest an alternative way to report results from features which overcome this weakness and provide the user with proven guaranteed accuracy. We also demonstrate that this method works as expected on a very hard dataset - the ImageNet challenge. It is known that density kernel methods fail in high dimensions. We show that our method works regardless. This is the novelty claim behind this suggestion.

---

### Meta-Review · Area_Chair1 · 2018-12-14

**Confidence:** 3
**Recommendation:** Reject

**Metareview:**

The paper presents a conformal prediction approach to supervised classification, with the goal of reducing the overconfidence of standard soft-max learning techniques. The proposal is based on previously published methods, which are extended for use with deep learning predictors. Empirical evaluation suggests the proposal results in competitive performance. This work seems to be timely, and the topic is of interest to the community.

The reviewers and AC opinions were mixed, with reviewers either being unconvinced about the novelty of the proposed work or expressing issues about the strength of the empirical evidence supporting the claims. Additional experiments would significantly strengthen this submission.